# Recent six-year atmospheric CO<sub>2</sub> concentration at the summit of Mt. Fuji observed by a battery-powered CO<sub>2</sub> measurement system

Shohei Nomura<sup>1</sup>, Hitoshi Mukai<sup>1</sup>, Yukio Terao<sup>1</sup>, Toshinobu Machida<sup>1</sup> and Yukihiro Nojiri<sup>1,2</sup>

<sup>1</sup>Center for Global Environmental Research, National Institute for Environmental Studies, 16-2 Onogawa, Tsukuba, Ibaraki, 305-8506, Japan.

<sup>2</sup>Hirosaki University, Bunkyo-1, Hirosaki, Aomori, 036-8560, Japan.

Correspondence to: Shohei Nomura (nomura.shohei@nies.go.jp)

# Abstract

5

We developed a battery-operated carbon dioxide (CO<sub>2</sub>) measurement system and successfully observed atmospheric 10 CO<sub>2</sub> concentrations at the summit of Mt. Fuji (3776 m a.s.l.) in Japan throughout the year since 2009, in spite of no power supply and severe low temperature over 10 months of a year. The observational results from 2009 to 2015 showed that CO<sub>2</sub> concentration at Mt. Fuji in summer and in winter was about 2–10 ppm lower and 2–12 ppm higher than at the Mauna Loa observatory (MLO), respectively. These episodic low concentrations at Mt. Fuji in summer have been cited as evidence that air masses originate from Siberia or China, which are affected by terrestrial CO<sub>2</sub> uptakes. The relatively higher

- 15 concentrations in winter were observed by air masses originated from China or Southeast Asia. The difference in monthly average CO<sub>2</sub> concentration between Mt. Fuji and MLO appeared to increase from 2009 to 2015. Interannual variability and growth rate of CO<sub>2</sub> concentration were similar both at Mt. Fuji and MLO, 13 ppm increase from 2009 to 2015, but the annual average concentration at Mt. Fuji was about 1 ppm higher than at MLO. Monthly averaged CO<sub>2</sub> concentration at Mt. Fuji exceeded 400 ppm in April 2013. Recent CO<sub>2</sub> concentration in 2015 at Mt. Fuji was about 62 ppm higher than the
- 20 previous record measured in 1980. To evaluate a regional representative of our measurement data, CO<sub>2</sub> values observed at Mt. Fuji were compared with airborne observations. They showed very good agreement with each other, indicating that Mt. Fuji was a representative site at which to monitor CO<sub>2</sub> concentration in the mid-latitude Asian region.

#### Keywords

25 Mt. Fuji, Battery-operated CO<sub>2</sub> measurement system, 100 batteries, Atmospheric CO<sub>2</sub> concentration, mid-latitude Asian region.

5

# **1** Introduction

To understand the global carbon cycle quantitatively, long-term observations of atmospheric  $CO_2$  concentration have been performed at "background sites" where the observed air is less-affected by local sinks and sources, such as Mauna Loa in Hawaii and Antarctica (Keeling et al., 1989; Conway et al., 1994; Keeling and Piper 2001). On the other hand, the importance of observations at "regional sites" where the air is influenced by regional sinks and sources has also been recognized, when analyzing regional sinks and sources using an inverse model with data from various ground observation sites such as Siberia and China (Maksyutov et al., 2003; Saeki et al., 2013; Zhang et al., 2014).

Globally, atmospheric CO<sub>2</sub> concentration is currently observed at about 180 sites, including both background and regional sites, although their distribution is not uniform (WMO, 2015). For example, in Southeast and East Asia, regions in which countries' economies have grown rapidly in this decade. In particular, China is now known as the country with the greatest CO<sub>2</sub> emissions in the world (CDIAC data (Boden et al., 2015)); But this region has only limited observation sites. Therefore, more observation sites are needed. Representative sites for the long-term observation of atmospheric CO<sub>2</sub> concentration in Japan are located in the higher and lower latitude coastal areas of Ochi-ishi, Minami-torishima Is., Hateruma Is., and Yonaguni Is. (Fig. 1) (Watanabe et al., 2000; Mukai et al., 2001).

Mid-latitude regions of Japan are most affected by regional air mass transportation via the prevailing westerlies from the Asian continent. However, suitable  $CO_2$  observation sites are lacking because of the distribution of industrial and populated areas. Obtaining  $CO_2$  observations in the mid-latitude Asian region would be advantageous for monitoring the change in the carbon cycle, because many countries in this region are growing economically with associated increases in their  $CO_2$  emissions.

- The summit of Mt. Fuji, which is the highest mountain in Japan (3776 m a.s.l.), could be considered a good site for monitoring CO<sub>2</sub> concentration. It is located in the free troposphere, which means it is not affected directly by air at ground level for most of the year (Tsutsumi et al., 1994). Mt. Fuji is positioned in the center of Japan (Fig. 1) and the air mass that passes over Mt. Fuji is mainly affected by the regional air characteristics of the Asian continent. Nakazawa et al. (1984) performed observations of CO<sub>2</sub> concentration at the summit of Mt. Fuji in October 1980 and July–October 1981, and they
- made a number of findings. (i) CO<sub>2</sub> concentration was not influenced by wind direction. (ii) Diurnal variation of CO<sub>2</sub> concentration was not observed. (iii) CO<sub>2</sub> concentrations observed on Mt. Fuji were in close agreement with several vertical profiles of CO<sub>2</sub> concentration derived from aircraft measurements near Sendai, Japan. (iv) Irregular CO<sub>2</sub> variations were caused by air masses transported from different regions such as Pacific Ocean and Asian Continent. (v) The observed annual rate of increase was comparable with the rate derived from aircraft measurements over Japan. Therefore, it was suggested
- that observations obtained at the summit of Mt. Fuji could be considered representative of the tropospheric  $CO_2$  concentration in the mid-latitude Asian region. Sawa et al. (2005) performed continuous observations of  $CO_2$  and CO concentration at the summit of Mt. Fuji from September 2002 to February 2003 and from May 2003 to May 2004, and they

5

found the followings: (i) many episodic events with large enhancements of  $CO_2$  concentration and (ii) the episodic enhancements were clearly associated with increased CO concentration peaks observed at the same time.

After 2004, the observation of CO<sub>2</sub> concentration at the summit of Mt. Fuji was interrupted because of the shutdown of the manual operation at the Mt. Fuji weather station by Japan Meteorological Agency (JMA). Although the JMA continued automatic meteorological observations, commercial power is only available for use during two months in the summer. Therefore, it became difficult to maintain whole-year observations at the station without a power supply and an air conditioner.

In this study, we developed a  $CO_2$  measurement system that can be operated without electricity supply, even the under harsh conditions found at the summit of Mt. Fuji, where the atmospheric pressure is low and the weather can be extremely 10 cold. Our system was insulated and developed to use battery power for over 10 months of a year. To minimize power consumption, our system was configured to measure  $CO_2$  concentration only at a certain hour in a day. To maintain longterm observations, the operational system included an auto-charging system for 100 batteries and a satellite communication system. Since 2009, we have successfully obtained  $CO_2$  data using this system at the station on Mt. Fuji.

In this paper, we evaluate the concentration level and trend of  $CO_2$ . Furthermore, to characterize recent  $CO_2$  concentration variations over Japan, the seasonal and daily variations of  $CO_2$  concentration are investigated in relation to air masses by 15 comparison with CO<sub>2</sub> data from the Mauna Loa observatory (MLO), Hawaii. To evaluate the regional representativeness and precision of the measurements obtained by our system, the data are compared with aircraft observations.

#### 2 Methods

#### 2.1 Location

- 20 Mt. Fuji is the highest mountain in Japan, located in the middle of mainland Japan (35.21°N, 138.43°E) on the Pacific side (Fig. 1). Mt. Fuji has not erupted since 1707 and no gas emission from the crater has been observed at the mountain summit. There is small outer rim (200-m height, 800-m diameter) surrounding the crater. The first Mt. Fuji weather station was constructed on the edge of the highest outer rim at Kengamine in 1936. In 1970, the old building was replaced by the present buildings. Until October 2004, several officers of the JMA worked throughout the year at the station for weather
- 25 observation.

At the summit of Mt. Fuji, the coldest daily temperature observed is -35°C in February 1981. The maximum daily temperature is about  $17^{\circ}$ C in August. During the passage of a typhoon, wind speed can peak at >70 m s<sup>-1</sup>. Annual average wind speed is about 12 m s<sup>-1</sup>.

As mentioned before, manual observations were halted and some weather observations were automated in 2004. A commercial electricity supply is provided only in summer because there are no workers on site from the end of August to the 30 following beginning of July. Our measurement system was installed in the 3<sup>rd</sup> building of the station in 2009 (Fig. 2).

#### 2.2 Measurement system

Figure 3(a)–(c) shows a photo of the CO<sub>2</sub> measurement system, a schematic of both the gas and the electricity flows, and the insulation method of the main measurement component, respectively. The system mainly consists of the main measurement component, power control devices for battery charging and power switching, and 100 shielded lead acid
batteries (12 V) (G42EP: Enersys Co. Ltd.; temperature range from –40 to 80°C). The main measurement component (CO<sub>2</sub>-MTF1, Kimoto Electric Co., Ltd.) includes NDIR (Li-840, LI-COR Co. Ltd.), air pumps (CM-15-12: E. M. P-Japan Ltd.; 1.2W), a drying system using membrane (Flemion: SWG A01-18: AGC Engineering Co. Ltd.) and a desiccant (Silica gel; 2000 ml). Each device in the measurement component was selected for minimal electricity consumption because the system operates using batteries from the end of August to the following the beginning of July.

- The main measurement component  $(30 \times 30 \times 30 \text{ cm})$  was covered with foam insulator (Phenovaboard: Sekisui Chemical Co. Ltd.) and placed in a 100-L plastic insulated box (RPS-100NF: REMACOM Co., Ltd.), which was also covered with 15-cm-thick Phenobaboad (Fig. 3(c)). The temperature at the summit of Mt. Fuji drops below  $-20^{\circ}$ C in winter. The control circuit boards had been tested under such conditions, the Li-840 sensor requires a certain temperature (50°C) when starting the measurements. Furthermore, the diaphragm rubber (Chloroprene) of the air pumps could be damaged in
- such low working temperatures. Therefore, the main measurement component was specially insulated to maintain a suitable working temperature and to save energy for starting the Li-840. In addition, the temperature of measurement component fall below 0°C, a small internal heater was planned to activate to maintain the temperature above 0°C when the pumps started. Each battery was wrapped in plastic film and placed in a corrugated cardboard box, which itself was wrapped in plastic film in case of liquid leakage and for heat insulation.
- An air inlet was placed along a water tank (about  $3 \times 4 \times 3$  m (D × L × H)), which was located just next to the 3<sup>rd</sup> building of the station (Fig. 2). The water tank was covered partly by wooden boards to be protected from snow but with sufficient space for air to penetrate between the wooden cover and the water tank. If the air inlet was placed completely outside, it would be difficult to take the air in winter because snow will cover the whole inlet. Air can be drawn through about 20 m of 1/4 inch PTFE tube by an air pump.
- A data communication antenna (AT1621-142W-THCF- 000-00-00-NM: AeroAntenna Technology Inc., Chatsworth, CA. U.S.A.) for the Iridium satellite (Iridium 9601: KDDI Co. Ltd.; 7.5 W max.) was also placed in the small space in wooden cover.

We constructed two sets of the main measurement component, which were switched in the operational system each summer. After one-year of operation, the pumps and other equipment were replaced in readiness for its next deployment.

# 30 2.3 Electrical power system

Because commercial power is only available from the beginning of July to the end of August, a battery system is required for operation in about 10 months. One hundred batteries (12 V) were connected in parallel and the voltage was

checked when the measurement system started. If the voltage was below 10.5 V, the system halted operation. The electrical power system has a device to switch from "winter mode" to "summer mode." In winter, we used the 100 batteries for powering the system, whereas in summer, we used another battery connected to a battery charger powered via the commercial supply.

- In summer (July–August), the 100 batteries (two series of 50 batteries) are charged by two specially designed charging controllers. After the battery connection is changed from "power mode" to "charge mode," each controller starts charging a pair of batteries. Normally, each controller can charge 50 batteries, one pair after another, over a 3-week period. In Fig. 3(b), the thin lines and arrows show the direction of electric current in the summer mode, and the thick lines and arrows show it in the winter mode.
- 10

5

At the end of August, the power system was switched to winter mode and the system operated using the 100 batteries until next July.

#### 2.4 Measurement sequence

Because of the limited power supply during winter, the measurement system was configured to operate for only about 3.5 hours per day. In order to monitor background levels of CO<sub>2</sub> concentration in the mid-latitudes over Japan, we initially selected the period 14:00–17:28 Japan standard time (JST) from July 20, 2009 to July 19, 2010. However, we subsequently changed the operational time to 21:00–00:28 JST, to avoid local daytime influences from transportation of the air mass around Mt. Fuji that might affect the CO<sub>2</sub> concentration over the summit of Mt. Fuji, which is similar to how observations are obtained at MLO.

- The measurement sequence is summarized in Fig. 4. Briefly, at 21:00 JST, the temperature of the measurement system 20 is checked and the heater operated for an hour if the temperature is less than 0°C. The Li-840 starts at 21:30 JST and the cell temperature increases to about 50°C within 30 min (14 W max). Then, from 22:00 JST, room air is introduced to the Li-840 (3.6 W) for 2 min using a small pump with a flow rate of 50 ml min<sup>-1</sup>. This process is intended to stabilize the flow line. Outside air is drawn into the system by a medium volume pump at a flow rate of 1.5 L min<sup>-1</sup>. An aliquot of just 50 ml min<sup>-1</sup> is introduced to the drying and measurement system under the direction of a mass flow controller for 8 min. Then, three
- 25 standard gases (about 360, 390, and 420 ppm), prepared by the Japan Fine Products Co. and calibrated against the NIES09 CO<sub>2</sub> scale (Machida et al., 2009), are measured for 4 min to calibrate the system. Generally, this sequence is repeated four times. From 23:28 JST, the Iridium satellite data communication is worked for one hour until the data are sent successfully. The derived concentration was based on the average of the data from the second, third, and fourth cycles, whose standard deviations was within approximately 0.2 ppm (Fig. 5).

# 30 2.5 Continuous measurement in summer

In order to measure the daily variation of  $CO_2$  concentration, a simple continuous  $CO_2$  measurement system was used in the summers of 2012 and 2013. It comprises the Li-840, flow-adjusting parts, a gas selector, and four standard gases. This

system has no drying component but the dry air base  $CO_2$  mole fraction is calculated from the H<sub>2</sub>O concentration by the Li-840. The precision of the measurement of  $CO_2$  concentration by this system is calculated to be about 0.3 ppm by preliminary examination.

# 2.6 Weather data

5 The temperatures in the room and inside the measurement system were monitored when the system measure CO<sub>2</sub> concentration in outside air. The external temperature at the summit of Mt. Fuji was measured by the JMA and the data were taken from the JMA website (http://www.data.jma.go.jp/obd/stats/etrn/index.php).

# 2.7 Backward air trajectory analysis and CO2 data to be compared to those of Mt. Fuji

Backward air trajectories were calculated using the METEX system (Zeng and Fujinuma, 2004) available via the 10 website of the Center for Global Environmental Research, National Institute for Environmental Studies (<u>http://db.cger.nies.go.jp/metex/trajectoy.jp.html</u>). They used vertical wind speed (z-direction) from the ECMWF 0.5 × 0.5 mesh data to calculate 72-h trajectories.

For comparison with the CO<sub>2</sub> data from the MLO, which is located as a similar altitude (3397 m a.s.l.; 19.54°N, 155.58°W), daily data (Tans and Keeling, 2016) were used (www.esrl.noaa.gov/gmd/ccgg/trends/). In addition, data from

15 aircraft measurements obtained via the CONTRAIL (Comprehensive Observation Network for Trace gases by Airliner) project (Machida et al., 2008) were also used. We selected CONTRAIL data over Japan in the region 34–36°N, 136–141°E at an altitude of 3600–3900 m during 2009–2013.

The trend of  $CO_2$  concentration was calculated according to the method of Thoning et al. (1989) with the cut-off frequency of 667 days (0.5472 cycles yr<sup>-1</sup>) for a Fast Fourier Transform filter.

# 20 3 Results and discussion

# 3.1 Operation with 100 batteries over 6 years

The total electric energy consumption of the  $CO_2$  measurement system was estimated to be about 3A for three hours per day. However, if a single battery can last 42 Ah, then 100 batteries could operate for 467 days under ideal conditions. As mentioned earlier, however, the ambient temperature on Mt. Fuji is low in winter and therefore, battery capacity might be reduced. In addition, the operation of the small heater in the system would shorten the duration of operation further.

25

Figure 6 shows the annual variations of the outside temperature on Mt. Fuji, in the observation room and in the measurement system protected by the insulation box. The outside temperature was very low until April, sometimes below – 20°C. The average temperature in January was  $-19.8 \pm 4.4$ °C. The room temperature ( $-15.4 \pm 1.9$ °C) was about 5°C higher than the outside temperature.

5

The temperatures of the batteries were almost similar to the ambient temperature, which would have affected their capacities. On the other hand, the temperature inside the insulation box was  $15^{\circ}$ C higher than the room temperature. Even in winter, the average temperature of the system was just above  $0^{\circ}$ C because small amounts of heat were produced by each device (e.g., the pump, circuit board, and Li-840 sensor) during the operational time, which was retained within the insulated box. Consequently, the system's small heater operated rarely in winter. The CO<sub>2</sub> measurement system was able to measure CO<sub>2</sub> concentration, such as the standard deviations of standard gas was within approximately 0.2 ppm (Fig. 5), stably and

precisely in spite of the low temperature condition.

Figure 7 shows the voltage of the batteries during the operation. The voltage just after charging in summer was 13.4– 13.6 V, but the voltage decreased gradually to 12.0–12.2 V by July of the following year. After March, the voltage seemed to recover as the temperature increased toward summer. Over the six-year period, voltages less than 10.5 V were not observed. Therefore, measurements of CO<sub>2</sub> concentration were not interrupted by power shortages. Thus, we concluded that our CO<sub>2</sub> measurement system, developed for the specific conditions of Mt. Fuji with its harsh weather and limited power supply, operated successfully for six years with only planned summer maintenance required.

On two occasions, the operation of the system was interrupted by damage to a power board in the main control unit 15 because of lightning (April 2–July 23, 2012 and August 1–18, 2014). In the summer season particularly, there are often thunderstorms observed at the summit of Mt. Fuji, because the station is in the cloud layer. Recently, a tight connection of the earth line between the observation building and the water tank facility was made to reduce the risk of lightning passing through the Iridium antenna cable. However, we are still unsure whether we can completely negate the risk of a lightning strike affecting the measurement system.

#### 20 3.2 Daily variation of CO<sub>2</sub> concentration

The  $CO_2$  concentrations at the summit of Mt. Fuji were observed continuously in the summers (July–August) of 2012 and 2013. Figure 8 shows the difference between the hourly and daily averages of  $CO_2$  concentration during the observed periods. The hourly average of the difference showed random variations within a range of about 0.5 ppm with no clear diurnal variation either in 2012 or in 2013. Therefore, the timing of the air sampling did not appear to affect the monthly and variation either in 2012 or in 2013.

25 yearly averages or the trend analysis.

Nakazawa et al. (1984) and Sawa et al. (2005) also reported that daily variation of  $CO_2$  concentration at the summit of Mt. Fuji was very small. Tsutsumi et al. (1994) observed  $O_3$  concentration at the summit of Mt. Fuji and they reported that the small daily variation observed was influenced by a mountain–valley wind system in summer; however, the variation was much smaller than at MLO, Niwot Ridge in the U.S.A., and Hakkouda in Japan. Igarashi et al. (2004) observed  $SO_2$ 

30

variation at the summit of Mt. Fuji, and although they did not find any daily variation, they did reveal episodic large-distance transport from the Asian continent. Although the sampling time was changed from daytime to night-time in July 2010, it is considered that such a change would not affect the results of the trend analysis.

According to observations at the summit of Mt. Fuji by Sawa (2005), the short-term cycle of CO concentration variation was related to neither wind direction nor wind velocity; however, its variation has been considered affected by regional-scale pollution over Asia.

Mt. Fuji is an independent peak, isolated from other mountain ranges. There is no vegetation above 2500 m, and because the station is located on the small outer rim (about 800-m diameter) of the crater at the summit, a strong wind always blows. However, the elevation of an observing station must be sufficiently high to minimize surface influences from local pollution and vegetation, which is why the summit of Mt. Fuji is considered a very suitable location for monitoring the background  $CO_2$  concentration over mainland Japan.

#### 3.3 Seasonal variation of CO<sub>2</sub> concentration

Observations of CO<sub>2</sub> concentration measured at the summit of Mt. Fuji (2009–2015) are plotted with daily data from CONTRAIL and MLO in Fig. 9. It was evident that a seasonal minimum occurs in September and that a maximum occurs in April or May; however, local minimum concentrations (episodic low-concentration) were often seen in July. The seasonal amplitude of variation at Mt. Fuji was same as the amplitude at CONTRAIL and about 18 ppm higher than the amplitude at MLO. In general, the seasonal amplitude of CO<sub>2</sub> variation is greater over land area than oceanic areas because of the 15 influence of photosynthesis, respiration of local vegetation, and anthropogenic CO<sub>2</sub> emissions.

Monthly averages of  $CO_2$  concentration at the summit of Mt. Fuji are presented in Table 1. The  $CO_2$  concentration at Mt. Fuji exceeded 400 ppm in April 2013, which was one year earlier than at MLO. The annual average of  $CO_2$  concentration at Mt. Fuji also exceeded 400 ppm in 2015. Japanese islands have a seasonal wind system (i.e., the monsoon), and the  $CO_2$  concentration must be affected by the wind direction and the origin of the air mass. To clarify the characteristics

- of the seasonal variation of  $CO_2$  concentration at Mt. Fuji, the difference between the daily concentration ( $\Delta CO_2$ ) at Mt. Fuji and MLO is plotted in Fig. 10. The  $CO_2$  concentration levels at both stations were similar in July to September, when oceanic air prevails over Japan, although occasional lower concentrations occurred in July at Mt. Fuji. Conversely, the  $CO_2$ concentration from December to March at Mt. Fuji was generally higher than at MLO.
- To clarify the difference between the two sites, a trajectory analysis was performed for the daily data obtained at Mt. 25 Fuji. We divided the surrounding region into five areas of air mass origin according to a 72-h trajectory analysis, as shown in Fig. 11. The ΔCO<sub>2</sub> concentrations for June–August and January–March are shown in Fig. 12(a) and 12(b), respectively, for the five areas of origin. It was found that the Siberian air mass always carries lower concentrations than the background air. In addition, Chinese air sometimes had much lower concentration than Siberian air (i.e., up to 10 ppm less), but it also sometimes had concentrations up to 7 ppm higher than at MLO. Such characteristics might be explained by CO<sub>2</sub> uptake by
- the vegetation of Siberia and China and by  $CO_2$  addition from anthropogenic emissions over the Asian continent. Conversely, air originating from the Pacific Ocean and the areas near Japan and Southeast Asia showed similar concentrations to MLO. Therefore, it was concluded that the reason why  $CO_2$  concentrations in summer were much lower than those of MLO was related to the origin of the air mass, especially when it was Siberia and China which are influenced by terrestrial ecosystem.

In winter, air from all areas except the Pacific Ocean showed higher concentrations than at MLO, indicating some influence from the continent, even Southeast Asia. In particular, air originating from China sometimes showed much higher concentrations than air from other areas, suggesting that anthropogenic  $CO_2$  might be added to the air over China in winter.

- Sawa et al. (2005) reported that CO<sub>2</sub> concentration at Mt. Fuji was 2 ppm lower than the level at Minami-torishima
  Island in summer but 3 ppm higher in winter. Minami-torishima Island is located 2000 km southeast of Mt. Fuji and 5000 km west of MLO. They stated that the air mass over Mt. Fuji must be influenced by continental air. Similarly, at Hateruma Island, which is much closer to China, the influence from the continent was clearly observed, especially in winter (e.g., Tohjima et al., 2010).
- The seasonal variation of CO<sub>2</sub> concentration at Mt. Fuji was characterized by much lower values than associated with 10 oceanic air, which is carried from Siberia and the Asian continent in summer and relatively higher concentrations than associated with Pacific air, which is influenced by emissions over the Asian Continent in winter. Because of the relatively higher concentrations from autumn to the following spring, annual CO<sub>2</sub> concentrations at Mt. Fuji during 2009–2015 were about 1 ppm higher than at MLO (Fig. 13).

#### 3.4 Trend analysis

- The CO<sub>2</sub> data obtained at the summit of Mt. Fuji by Nakazawa et al. (1984) in 1980-1981, Sawa et al. (2005) in 2003-2004, and in this study are plotted in Fig. 14, together with data from MLO. The CO<sub>2</sub> concentration increased to as high as 62 ppm over the 35 years from 1980 to 2015. The annual rate of increase during 2009–2015 was 1.5–2.7 ppm yr<sup>-1</sup> (Fig. 13). This was because anthropogenic CO<sub>2</sub> emissions have increased to >10 Gt-C yr<sup>-1</sup> in recent years, while they were only as high as 5 Gt-C yr<sup>-1</sup> in the 1980s (Le Quéré et al., 2015). In particular, higher growth rates were observed in 2012. In general, El Nino years lead to a higher global surface temperature, which is connected to the increased rate of growth of CO<sub>2</sub> concentration because of accelerated plant respiration over land and weakened photosynthesis activity. However, 2012 was not an El Nino year; therefore, this year is a special type of year. This recent growth rate was almost the same as at MLO, suggesting that the long-term stability for our CO<sub>2</sub> observation system including standard gases is reliable.
- The seasonal trend of CO<sub>2</sub> concentration also revealed some interesting characteristics. For example, the negative values of  $\Delta$ CO<sub>2</sub> concentration have enlarged gradually over the six-year period of 2009–2014, as shown in Fig. 10. Chen et al. (2014) reported that growth of Asian vegetation increased recently. In particular, remote sensing observations over eastern Siberia have revealed a notable increase in vegetation during recent decades. Correspondingly, the negative values of  $\Delta$ CO<sub>2</sub> concentration have enhanced gradually. On the other hand, the relatively higher  $\Delta$ CO<sub>2</sub> concentration peaks in winter have gradually increased over the six years. This phenomenon might be related to the increase of anthropogenic CO<sub>2</sub>
- 30 emissions in China during recent decades (CDIAC database (Boden et al., 2015)). Such event-based phenomena should be evaluated by numerical simulation, but we expected that such signals from regional emissions or absorption changes would be included in the data at Mt. Fuji.