# Peer review of "Recent six-year atmospheric CO2 concentration at the summit of Mt. Fuji observed by a battery-powered CO2 measurement system"

_Atmospheric Measurement Techniques, 2016_

## Referee Comment (RC1) · Anonymous Referee #1 · 11 Nov 2016

GENERAL COMMENTS

The paper present a new proof of concept and interesting application of a system for the execution of long-term continuous CO2 measurements to be used in a remote, unmanned measurement site characterized by the total absence (for several months) of electric power. The authors provided enough technical details on the proposed set-up to allow reproduction at other measurement sites. Moreover, new data were presented about CO2 variability at Mt. Fuji (Japan).

My first general comment is that I'm not completely sure that this paper by Nomura et al., is completely fitting to AMT. As specified by AMT web site, AMT is dedicated to the" publication and discussion of advances in remote sensing, as well as in situ and

laboratory measurement techniques for the constituents and properties of the Earth's atmosphere. The main subject areas comprise the development, intercomparison, and validation of measurement instruments and techniques of data processing and information retrieval for gases, aerosols, and clouds." The paper by Nomura et al. presented an automatic, low-consumption system (even if based on already well-known and used commercial system), for the execution of $CO_2$ measurements at Mt. Fuji or at other remote, unmanned, adverse-weather measurement sites. Thus, for this aspect, it fits without any doubt to AMT. However, my feeling is that the paper is too unbalanced towards the analysis and exploitation of the $CO_2$ data series provided by this system or in demonstrating the geographical representativeness of the measurement site (which is more fitting to ACP in my view), rather than (as should be done for AMT, I think) in presenting and commenting the experimental set-up or in demonstrating/assessing the quality and the strong/weak points of the presented system.

Another major point is that the authors did not provide enough information to allow a robust assessment of the system performance and, thus, possible application. I think that the scientific purposes of this set-up should be better addressed. As an instance, I'm wondering if this set-up (based on the NDIR LI-COR Li-840) can be able to meet the data compatibility goal (0.10 ppm) requested by WMO for long-term global $CO_2$ measurements...(I'm skeptical about this).

Also the abstract should provide more details/information about the technical set-up and about the results of possible QA/QC test used to assess the quality of the data provided by the proposed set-up. Now, the abstract mostly provides a series of conventional atmospheric data analysis results, which (even if interesting) are not, I guess, the main topic for AMT.

I suggest that the authors can make more use of supplementary material (SM). Too many figures are presented in the manuscript up to now (17!). Some of them (e.g. Figure 2 and others – see my specific comments) can be moved to SM or skipped. The title clearly reflect the contents of the paper, but I would suggest recommend to modify keywords. I would suggest: carbon dioxide, unattended system, low-consumption system, Japan, long-term observations

By the way, several points should be addressed before publication (several of them are also related to assess the "performance" of this new set-up). Moreover, along the paper some inaccurate terms are used (e.g. "precision") and some analyses are too "qualitative" (i.e. lacking of robust statistic).

At some place in the paper, the authors should mention the exact period of operation of the measurements, i.e. "month 2009 – month 2015" (this info is only present in the abstract).

Thus, for summarizing, I think that the paper can be considered for AMT, but only after major revisions. Basically, I suggest to the authors to strength the analysis of the performance of their system, to clearly assess the typical application for which this system is suitable for (i.e. long-term monitoring – if yes which data quality objective can be reached- vs air quality vs event studies) and to consider the comparison with Mauna Loa time series as a "case study" or a "typical application" for the proposed system.

SPECIFIC COMMENTS

Concerning the experimental set-up, I strongly recommend to change the PTEFE with stainless steel or Synflex 1300 tubing (PTFE is well known to be not optimal for CO2 sampling). By which frequency did you change the PTFE tubing? I would like to see more information about the working standards for routinely calibration of the system (i.e. flask material, mixture matrix, pressure regulators) and more discussion about the choice of your calibration strategy (are 4 min sufficient enough to obtain stable measurements?).

Pag 3, line 16. You cannot use this comparison of data for evaluating the precision (i.e. the repeatability) of your measurements! I also have serious concern that you can

provide an assessment of spatial representativeness by using this data-set...

I think that the authors should spent more work in presenting an assessment of the quality of the system performance. Along the manuscript, the authors provided a statistical analysis (Fig. 5 of the manuscript) of the standard deviation of the measured value of standard gases. This information can be useful to assess the "long-term repeatability" of the measurements (see e.g., Lebegue et al., AMT, 2016). However, to better exploit this point, I would like to see this analysis also split as a function of the different standard gases used and/or as a function of the different years. It should be interesting to see the absolute difference between the CO2 measured values of the standard gases and the reference standard values. Moreover I would like to see information about the uncertainty or expanded uncertainty of the measurements achievable by this system, which is an important parameter missed in the paper.

Concerning the difference with MLO, which may be the role of using two calibration scales (NOAA vs NIES09)? Please quantify... How many often did you re-calibrate the working standard against your National reference (I suppose once per year, but this should be clearly stated in the paper)?

Another important information to provide is the data coverage (%) over the measurement period: I feel this is rather high, but this represent a parameter (you must mention it along the abstract!) to evaluate the success of the presented system.

Pag 6, line 2: " The precision....is calculated to be about 0.3ppm". As recommended by the WMO/GAW glossary, "precision" is a term to be used only in relative terms. How did you calculate this "precision"? If this is a standard deviation, it is the "repeatability" or the "reproducibility". Any information about the "uncertainty" of the measurement?

Pag 7, line 24. For year 2012 it seems that the CO2 variability (error bars) experienced a diurnal behavior (higher variability during night-time): please comments. Does a diurnal cycle exist for internal temperature or other system parameters that can affect the measurements? Figure 8: you did not provide explanation in the Figure caption

for the error bars (please, do it). To investigate if statistical significant diurnal variability exist, you should plot for each hour an average mean value with the 95% confidence interval (not simply the standard deviation).

Pag 8, line 6: "...a strong wind always blow". Too generic. Please provide data or references.

Pag 8, line 11: "it was evident that...often see in July". It is difficult from this plot to obtain information about monthly values. I would suggest to replace this plot with monthly mean values and related error bars (representing the associated 95% confidence interval). This would also allow to skip Table 1.

Pag 8, line 24: Since the $CO_2$ is characterized by a long atmospheric lifetime, 72h back-trajectories can explain just a fraction of its variability. Please comment in the paper. It is possible to plot the back-trajectories (as "number concentration field" or as centroids of the detected clusters) over the spatial domain in Figure 11? In any case, consider to move Figure 11 to SM. You should also provide the percentage of occurrence for the different air mass transport class. The discussion about results reported by Figure 12 is too qualitative. For the summer, it seems that a large amount of variability affect the data set. For each class of air mass transport please provide average values and 95% confidence interval. How did you define background air (Pag 8,line 27)?

Pag 9, line 1: In my opinion, a too low number of Pacific Ocean air-masses were observed during winter (at least basing on the visual inspection of Figure 12 since no information about their seasonal occurrences were provided) for provide any kind of comment. "air from China SOMETIMES...". Too qualitative, please be more specific! How many times (frequency) did you observed these spikes with air-mass from China? The sentence from line 9 to 12 is not clear. I think some words were missed somewhere...However, I do not think that this paragraph and figure 13 add really important information to the paper, thus they can be skipped.

**[AMTD](url)**

Interactive
comment

The comparison with CONTRAILS observations must be better commented (or even skipped). The authors claim for a good agreement but I do not completely agree: a rather large "average" bias (2 ppm) can be deduced by the linear correlation result. In general I do not think that you can use this comparison for "verify" the precision of the measurement system (the term "precision" is not correct at page 10, line 16)! Also concerning the assessment of the representativeness of the measurement site, I think that this analysis can provide only a "qualitative" information (also considering that CONTRAILS observations are not uniformly distributed on the geographical domain). As an instance, the most part of the CONTRAILS observations are recorded not far from Tokyo. Would this introduce a bias on the contrails data-set due to the urban emissions? Can this (at least) partially explain the 2 ppm bias? Moreover, it is not clear what do you mean for "daily data". For CONTRAILS what kind of daily average did you consider? 24h mean values? Or did you select the hourly data on a time window in agreement with Fuji measurements (i.e. 14:100-17:28 and 21:00-00:28)?

Fig. 14 is almost unreadable. Mt. Fuji points completely overlap the "reference" time series of MLO. I'm wondering if it's really necessary to show the whole long term time series of MLO: for the most part of it, you do not have any data to overlap!!! I would suggest to start the x-axis since 2010 and make the MLO point more visible.

Fig. 9, line 17: "The RANGE of the annual rates of increase during 2009-2015 was 1.5-2.7 ppm/yr". Actually, the variability in the growth rate can be also related to other factors, as an instance ENSO (as mentioned in the following by authors) or to change in fluxes between atmosphere and terrestrial biosphere.

Pag 10, discussion of Fig. 15. Some statistical analysis should be provided for the linear correlation (i.e. statistical significance). In general, I think that is dangerous to compare and comment (small) long-term $CO_2$ differences between two measurement sites without providing an assessment of the measurement compatibility...I'm wondering if these tendency in long-term differences are evident also at other measurement sites. In this case, this would reinforce your conclusions. I recommend to consider

open-access data-sets available at the World Data Center for Greenhouse Gases by WMO/GAW and to repeat a similar analysis for other sites in the East Asia/Pacific regions vs MLO.

TECHNICALS: Pag 2 line 10: "(CDIAC, Boden et al., 2015)" line 15: "air mass transport" line 23: "Mt Fuji is positioned. . ..of the Asian continent". Please provide references or show back-trajectory analysis. Line 22: "Mt. Fuji is located. . ." Line 27: What do you mean for "irregular"? Please use a more specific terminology.

Pag3, Line 28, point (v): it is difficult to obtain information on the annual rate of increase basing on so "sparse" measurement.

Pag 4 Line 2: "shows a picture. . ." Line 3: "The system consists. . ." Line 16: "In addition, WHEN the temperature. . ."

Pag 8 Line 27:" It was found,. . .IN SUMMER"

---

## Referee Comment (RC2) · Anonymous Referee #2 · 11 Nov 2016

Title: Recent six-year atmospheric CO2 concentration at the summit of Mt. Fuji observed by a battery-powered CO2 measurement system

Authors: Shohei Nomura et al.

General comments:

The manuscript presents a technical description of a CO2 monitoring system operated for 6 years at the high altitude site Mt. Fuji, Japan, and associated data analysis. The description of the measurement setup is rather short for a publication in Atmospheric Measurement Techniques and should be extended (see specific comments below). On the other hand, the data discussion could be shortened as it is not the main scope of

a paper in AMT. This would be easily possible as the current manuscript extensively compares the Mt. Fuji data with data from Mauna Loa, Hawaii (differences of daily averages (Fig. 10), differences of daily averages for summer and winter (Fig. 12), trends and growth rates for Mt. Fuji and Mauna Loa (Fig. 13), Mt. Fuji and Mauna Loa time series (Fig. 14), trends of monthly Mt. Fuji-Mauna Loa differences (Fig. 15)). Alternatively, the authors can also envisage to elaborate the data interpretation section, e.g. by also incorporating observations from other (high altitude and/or background) monitoring stations, and can consider submitting the manuscript to Atmospheric Chemistry and Physics. Thus, I recommend the publication of the manuscript in AMT after strengthening the experimental section and concisely abridging the data interpretation. A revision of the manuscript with a stronger focus on the data analysis and a submission to ACP seems to be a suitable alternative, too.

The manuscript contains quite a lot of tables and figures, some of them are to my mind not really needed or can be merged. The authors may consider reducing the number of tables and figures, see also my comments below.

I strongly suggest revising the conclusions which are currently only a summary of what was said before. Please add the lessons-learnt (especially on the instrumental side) – e.g. what would you do differently when you could start from scratch again – and provide an outlook, e.g. are the measurements ongoing? If so, are there any changes/improvements on the measurement setup planned? Did the authors modify the measurement setup during the 6 years of operation, i.e. did they improve their system based on the experience gained in the earlier years?

Even if it is common to use expressions such as "the CO2 concentration was 400 ppm" colloquially, it is an incorrect statement and not suitable to be used in the scientific literature. Quantities given in ppm refer to mole fractions or mixing ratios and cannot be called concentrations. Please use the correct terminology throughout the manuscript.

Specific comments:

Title:

I suggest changing the title to "Six-years of atmospheric CO2 observations at Mt. Fuji recorded with a battery-powered measurement system"

Abstract:

Lines 15-18: "The difference in monthly average CO2 concentration between Mt. Fuji and MLO appeared to increase from 2009 to 2015. Interannual variability and growth rate of CO2 concentration were similar both at Mt. Fuji and MLO, 13 ppm increase from 2009 to 2015, but the annual average concentration at Mt. Fuji was about 1 ppm higher than at MLO." It sounds like a contradiction (the difference increases but the average was 1 ppm higher), or is at least not understandable without having seen Fig. 15 and the related discussion, respectively. I suggest to delete the first sentence and to add another sentence after the second sentence, like "However, differences in Mt. Fuji and MLO observations show divergent trends depending on seasons."

Line 18-19: delete "Monthly averaged . . . in April 2013." Not relevant here.

Lines 21 – 22: change ". . . indicating that Mt. Fuji was a representative site . . . in the mid-latitude Asian region" to ". . . indicating that Mt. Fuji is a representative site to monitor CO2 concentrations in the mid-latitude region."

Introduction:

Page 3, lines 8-9: ". . . without electricity supply . . .", better say ". . . without gridded electricity supply . . ."

Page 3, lines 8-9: reword sentence: ". . . even under the harsh conditions . . ."

Page 3, lines 8-9: " . . . harsh conditions found at the summit of Mt. Fuji, where the atmospheric pressure is low . . ." Why is low pressure a harsh condition?

Page 3, line 16: add latitude, longitude and altitude for Mauna Loa

Page 3, line 17: "To evaluate the regional representativeness and precision of the measurements obtained by our system, the data are compared with aircraft observations." Don't you evaluate the accuracy rather than the precision when comparing with other data?

Methods:

This should be the main part of the paper and thus, needs elaboration.

Page 3, lines 20-25: add Mount Fuji altitude. How can you access the station? Is access only possible in July and August, or also during the rest of the year (in exceptional cases, e.g. for trouble-shooting).

Page 4, line 6: add number of pumps (4, according to Fig. 3). Why do you need four pumps? Can you redesign the setup using less pumps reducing the power consumption?

Page 4, line 7: write "... using a Nafion membrane ...".What is the dew point that you achieve with the setup? Does the drying efficiency change (decrease) with time? Is a 2 litre cartridge of Silica gel sufficient for the Nafion counterflow for 10 months.

Page 4, first paragraph: did you apply any modifications to the measurement setup, in particular to the CO2 analyzer? E.g. disconnecting the display or reducing the flow through the NDIR to reduce power consumption.

Page 4, line 17: "... a small internal heater was planned to activate ..." was it only planned or was it also in place? If the latter is true, write "... a small internal heater was implemented to activate ..."

Page 4, lines 20-24: did you use any inlet filter? If so, how often was it changed?

Paragraph 2.3 Electrical power system:

It remains unclear why gridded power is only available in summer. What does exactly change in early July and late August? Is the station permanently staffed in summer?

Did you ever try off-grid generated power with wind turbines or solar panels?

Paragraph 2.4 Measurement sequence:

How is the measurement system controlled? Which software is used? Elaborate on the instrument maintenance. Can you remotely access the measurement setup, i.e. can you access the computer when the satellite communication is running? E.g. to modify the measurement sequence. Did you never face any serious instrumental failures? There seems to be a longer data gap in 2012 (according to Fig. 10)? What happened? What is the overall data coverage based on your daily averages?

What is the yearly consumption of reference gas? How long do the reference cylinders last?

Page 5, lines 15-18: "However, we subsequently changed the operational time to 21:00–00:28 JST, to avoid local daytime influences from transportation of the air mass around Mt. Fuji that might affect the $CO_2$ concentration over the summit of Mt. Fuji, which is similar to how observations are obtained at MLO." To my knowledge, $CO_2$ measurements at MLO are continuous and a filter to extract background conditions is applied afterwards. However, most background data are identified at Mauna Loa in the late afternoon, see http://www.esrl.noaa.gov/gmd/ccgg/about/co2_measurements.html for more details.

Page 5, line 22: did you use an inlet filter when measuring room air?

Page 5, line 22: "... to stabilize the flow line." What does that mean?

Page 5, lines 23-24: Why do you need to push the air into the analyzer when another pump sits behind?

Page 5, line 27: why does it need one hour to send the data? How large are the data files? Which time resolution do you store the $CO_2$ data?

Page 5, lines 28-29: "The derived concentration was based on the average of the data

from the second, third, and fourth cycles . . .", in other words, the daily average is based on 3 x 8 min = 24 minutes of observations. In fact, it will be even less since you have to discard some data to account for flushing and signal stabilization after switching from room air to outside air. Add the information how many minutes of data were discarded after switching. Please mention explicitly that the daily averages used below only rely on a very short measurement period.

Paragraph 2.5 Continuous measurements in summer:

Does the continuous system use the same inlet? Did you use the same reference gases? Does the default system operated all year long also only measure for 3.5 hours a day in summer? The summer system has no dryer. Did you test and quantify the $CO_2$ losses in the Nafion dryer?

Paragraph 2.6 Weather data:

How does the power supply for the meteorological measurements look like?

Page 7, lines 14-19: again, didn't you experience any other interruptions, in particular during the 10 months of unattended operation?

Paragraph 3.2 and onwards:

Use data other than Mauna Loa for comparison and interpretation. E.g. Lulin (Taiwan), Niwot Ridge (USA), Mt. Waliguan (China) etc.; use the marine boundary layer reference (available at http://www.esrl.noaa.gov/gmd/ccgg/mbl/) for comparison which is also available for the Mt. Fuji latitude. How does the difference in latitude (Mt. Fuji – Mauna Loa) influence your comparison?

Page 7, line 29: add altitudes for Niwot Ridge and Hakkouda

Page 8, lines 4 – 8: move this paragraph up to paragraph 2.1

Page 8, line 13: say "18 ppm larger", add amplitudes for Mt. Fuji and MLO.

Page 8, line 16: Table 1 is not needed, in particular if data are available in a publicly accessible data repository. Did you submit the data to the World Data Centre for Greenhouse Gases?

Page 8, lines 22 – 23: "Conversely, the CO2 concentration from December to March at Mt. Fuji was generally higher than at MLO." February and March are the moths with most intense biomass burning on the Indochinese Peninsula. Do these fires affect the observations at Mt. Fuji?

Page 9, line 5: add reference to Fig. 1

Page 9, lines 12-13: this statement is based on the Fourier-transformed (i.e deseasonalized) data, correct?

Page 9, lines 20- 21: "…the increased rate of growth of CO2 concentration because of accelerated plant respiration over land and weakened photosynthesis activity". Add reference. Next to vegetation effects, it is also due to more intense biomass burning, see e.g. Betts et al., Nature Climate Change, September 2016).

Page 9, lines 24-26: "For example, the negative values of ∆CO2 concentration have enlarged gradually over the six–year period of 2009–2014, as shown in Fig. 10. Chen et al. (2014) reported that growth of Asian vegetation increased recently." I doubt that such an effect can be seen in a 6-year time series. Moreover, Chen et al. refer to changes over the last three decades.

Page 9, lines 26-28: "In particular, remote sensing observations over eastern Siberia have revealed a notable increase in vegetation during recent decades. Correspondingly, the negative values of ∆CO2 concentration have enhanced gradually." Is there any proof confirming this statement. How about CO2 observations at other Asian sites? Can't it also be an emission effect with decreasing emissions in summer?

Page 9, lines 28-29 and page 10, lines 6-7: "This phenomenon might be related to the increase of anthropogenic CO2 emissions in China during recent decades" and

".. .the positive trend in winter was not so significant, which might be attributable to the slowing of the growth rate of CO2 emissions in China during 2011–2014." Sounds like a contradiction.

Page 9, lines 20-32: "Such event-based phenomena should be evaluated by numerical simulation, but we expected that such signals from regional emissions or absorption changes would be included in the data at Mt. Fuji." If you expected it, why didn't you look closer at it?

Conclusions:

As mentioned above, revise the conclusions and add lessons-learnt and an outlook.

Tables and Figures:

Table 1: not needed, a release of the data in a public data repository is strongly encouraged.

Table 2: not needed, some numbers could be incorporated in Fig. 15.

Fig. 1: add areas of air mass origin (Fig. 11) to Fig. 1 and delete Fig. 11.

Fig. 5: what is the nominal value of the standard gas?

Fig. 6, caption: Write "Monthly averages of ambient temperatures outside . . .."

Fig. 11: merge with Fig. 1 and delete

Fig. 14: does it show daily averages? Monthly averages? The figure repeats Fig. 9, the long-term evolution is not of real interest here. I suggest to delete it.

Fig. 15: only a few trend lines are plotted: for which months? Use open symbols for the months without trend line?

Fig. 16: add information to Fig. 1 and delete Fig. 16

Minor comments:

Page 2, line 11: start with lowercase letter after semicolon.

Page 5, line 12: typo "Phenobaboad"

---

## Author Comment (AC1) · 26 Dec 2016

**Response to reviewer #1 Recent six-year atmospheric $CO_2$ concentration at the summit of Mt. Fuji observed by a battery-powered $CO_2$ measurement system**

Shohei Nomura[a], Hitoshi Mukai[a], Yukio Terao[a], Toshinobu Machida[a] and Yukihiro Nojiri[a,b]

[a]Center for Global Environmental Research, National Institute for Environmental Studies, 16-2 Onogawa, Tsukuba, Ibaraki, 305-8506, Japan.

[b]Hirosaki University, Bunkyo-1, Hirosaki, Aomori, 036-8560, Japan.

GENERAL COMMENTS

1. The paper present a new proof of concept and interesting application of a system for the execution of long-term continuous CO2 measurements to be used in a remote, unmanned measurement site characterized by the total absence (for several months) of electric power. The authors provided enough technical details on the proposed set-up to allow reproduction at other measurement sites. Moreover, new data were presented about CO2 variability at Mt. Fuji (Japan).

   >Thank you very much for your explanation.

2. My first general comment is that I'm not completely sure that this paper by Nomura et al., is completely fitting to AMT. As specified by AMT web site, AMT is dedicated to the" publication and discussion of advances in remote sensing, as well as in situ and laboratory measurement techniques for the constituents and properties of the Earth's atmosphere. The main subject areas comprise the development, intercomparison, and validation of measurement instruments and techniques of data processing and information retrieval for gases, aerosols, and clouds." The paper by Nomura et al. presented an automatic, low-consumption system (even if based on already well-known and used commercial system), for the execution of CO2 measurements at Mt. Fuji or at other remote, unmanned, adverse-weather measurement sites. Thus, for this aspect, it fits without any doubt to AMT. However, my feeling is that the paper is too unbalanced towards the analysis and exploitation of the CO2 data series provided by this system or in demonstrating the geographical representativeness of the measurement site (which is more fitting to ACP in my view), rather than (as should be done for AMT, I think) in presenting and commenting the experimental set-up or in demonstrating/assessing the quality and the strong/weak points of the presented system.

   >We re-arranged the manuscript for AMT. We made one section for explanation of system performance and added other explanations. We added our calibration information about stability of working standards, their usage duration and their scale differences from NOAA, which was the WMO standard scale. Also, we added the data about comparison between measurement results of a battery-powered measurement system and bottle sampling. In addition, we omitted a part of measurement results in atmospheric $CO_2$ concentration and shorten the measurement results part.

3. Another major point is that the authors did not provide enough information to allow a robust assessment of the system performance and, thus, possible application. I think that the scientific purposes of this set-up should be better addressed. As an instance, I'm wondering if this set-up (based on the NDIR LI-COR Li-840) can be able to meet the data compatibility goal (0.10 ppm) requested by WMO for long-term global $CO_2$ measurements. . .(I'm skeptical about this).

>The electrical power supply at Mt. Fuji is limited. No gridded power supply from September to following June. Therefore, Li-840 was better to be used in terms of saving energy consumption. We averaged the signal from Li-840 for 2 min in the case of standard gases, a linearity of the calibration and its reproducibility were very good. In the new section (3.1) which we added, we made explanation for checking the linearity and the differences between the measured values and the assigned values of the standard gases. The differences from linear fit were smaller than 0.05 ppm, suggesting that the uncertainty of measured values is within 0.1ppm.

4. Also the abstract should provide more details/information about the technical set-up and about the results of possible QA/QC test used to assess the quality of the data provided by the proposed set-up. Now, the abstract mostly provides a series of conventional atmospheric data analysis results, which (even if interesting) are not, I guess, the main topic for AMT.

>We added the explanation about measurement system and QA/QC tests using two comparison works in the abstract. Also, we added the measurement performance for 6 years. According to your suggestion, we deleted some results of atmospheric data analysis.

5. I suggest that the authors can make more use of supplementary material (SM). Too many figures are presented in the manuscript up to now (17!). Some of them (e.g. Figure 2 and others – see my specific comments) can be moved to SM or skipped. The title clearly reflect the contents of the paper, but I would suggest recommend to modify keywords. I would suggest: carbon dioxide, unattended system, low-consumption system, Japan, long-term observations.

> We decreased number of figures to six.
Merged: Figure 1 and 11, and Figure 3 and 4, and Figure 9 and 13,
Created: one table and two figure, Move to supplementary material; Table 1).
    We changed the keywords based on the suggestion. But we would like to take "battery-powered system" instead of "unattended system", "low-energy consumption" instead of "low-consumption system", "Mt. Fuji" instead of "Japan", and "long-term monitoring" and. "Carbon dioxide"

6. By the way, several points should be addressed before publication (several of them are also related to assess the "performance" of this new set-up). Moreover, along the paper some inaccurate terms are used (e.g. "precision") and some analyses are too "qualitative" (i.e. lacking of robust statistic).

>We changed the term "precision" to "repeatability" or "accuracy" depending on the cases. We tried to change qualitative explanation to more quantitative, using data and statistic.

7. At some place in the paper, the authors should mention the exact period of operation of the measurements, i.e. "month 2009 – month 2015" (this info is only present in the abstract).

>We wrote the observation period (July 2009-December 2015) at the experimental section 2.4.

8.  Thus, for summarizing, I think that the paper can be considered for AMT, but only after major revisions. Basically, I suggest to the authors to strength the analysis of the performance of their system, to clearly assess the typical application for which this system is suitable for (i.e. long-term monitoring – if yes which data quality objective can be reached- vs air quality vs event studies) and to consider the comparison with Mauna Loa time series as a "case study" or a "typical application" for the proposed system.

> We made major revision, as suggested. To strengthen the explanation about analytical performance, we added one section (3.1 analytical performances) which included calibration information on working standard gas and its stability. We added one table for the CO2 concentration stability in the standard gases cylinders. Linearity and deviation the data from the calibration curve between the measured values and the assigned values were evaluated. We also showed a results of comparison between $CO_2$ values of the system and the bottle sampling experiment.

| Cylinder No | Before the cylinders installation | | | | After the cylinders replacement | | | | Change of concentration | |
| --- | --- | --- | --- | --- | --- | --- | --- | --- | --- | --- |
| | Date of calibration | Calibrated value (ppm) | Pressure of Cylinder (MPa) | Date of installation | Date of replacement | Date of re-calibration | Re-calibrated value (ppm) | Pressure of Cylinder (MPa) | Change amount of the concentration (ppm) | Change rate (ppm year$^{-1}$) |
| CPC-00449 | 17-Jun-2009 | 368.86 | 10.8 | 16-Jul-2009 | 24-Jul-2011 | 19-Aug-2011 | 368.82 | 3.4 | -0.03 | -0.02 |
| CPC-00447 | 17-Jun-2009 | 383.10 | 11.0 | 16-Jul-2009 | 24-Jul-2011 | 19-Aug-2011 | 383.24 | 4.0 | 0.14 | 0.06 |
| CPC-00448 | 17-Jun-2009 | 403.45 | 10.8 | 16-Jul-2009 | 24-Jul-2011 | 19-Aug-2011 | 403.54 | 3.2 | 0.09 | 0.04 |
| CPC-00445 | 10-Jun-2011 | 367.94 | 11.4 | 25-Jul-2011 | 25-Jul-2013 | 6-Aug-2013 | 368.02 | 0.8 | 0.09 | 0.04 |
| CPC-00450 | 10-Jun-2011 | 383.46 | 11.5 | 25-Jul-2011 | 25-Jul-2013 | 6-Aug-2013 | 383.42 | 3.6 | -0.04 | -0.02 |
| CPC-00451 | 10-Jun-2011 | 402.29 | 11.5 | 25-Jul-2011 | 25-Jul-2013 | 6-Aug-2013 | 402.37 | 3.4 | 0.08 | 0.04 |
| CPC-00043 | 23-Jun-2013 | 367.10 | 12.8 | 26-Jul-2013 | 1-Jul-2016 | 10-Jul-2016 | 367.12 | 2.5 | 0.02 | 0.01 |
| CPC-00448 | 23-Jun-2013 | 393.17 | 12.6 | 26-Jul-2013 | 1-Jul-2016 | 10-Jul-2016 | 393.12 | 2.5 | -0.05 | -0.02 |
| CPC-00449 | 23-Jun-2013 | 418.59 | 12.8 | 26-Jul-2013 | 1-Jul-2016 | 10-Jul-2016 | 418.44 | 2.5 | -0.15 | -0.05 |
| CPC-00445 | 15-Jun-2016 | 389.18 | 13.2 | 2-Jul-2016 | | | | | | |
| CPC-00450 | 15-Jun-2016 | 409.15 | 13.2 | 2-Jul-2016 | | | | | | |
| CPC-00451 | 15-Jun-2016 | 429.16 | 13.2 | 2-Jul-2016 | | | | | | |

[Figure]

[Figure]

[Figure]

[Figure]

SPECIFIC COMMENTS

9.  Concerning the experimental set-up, I strongly recommend to change the PTEFE with stainless steel or Synflex 1300 tubing (PTFE is well known to be not optimal for CO2 sampling). By which frequency did you change the PTFE tubing? I would like to see more information about the working standards for routinely calibration of the system (i.e. flask material, mixture matrix, pressure regulators) and more discussion about the choice of your calibration strategy (are 4 min sufficient enough to obtain stable measurements?).

> We wanted to use stainless steel tube or Synflex for inlet tube. But Mt. Fuji including the station is categorized as the National park. When the equipment install outside the station, the equipment was required to be colorless and transparent to conserve the landscape. So we chose the PTFE tube for the inlet tube, as a second choice. We did not replace the PTFE tube after the first installation at the station.   But we made leak check every summer. This detail was written in 2.2 measurement system.

We added table 1 which showed calibrated values, pressure and re-calibrated values for the standard gas cylinders. These standards were produced by Japan Fine Product Co. in aluminum 10L cylinder.   CO2 gas was diluted by purified natural air as zero air. Special regulators for cold environment use which were provided by Nissan Tanaka Co. were used. This detail was written in 2.4 measurement sequence.

We added one figure (Fig. 3) to show the typical NDIR signal for standard gas and outside air measurement. As shown in the figure, signal became stable 2 min after starting measurement. So, we discard the data for first 2 min and averaged data for the rest of the time (in the case of standard data was averaged for 2 min, in the case of the air data was averaged for 6 min.)   The measurement sequence ( 3 standards=> room air => outside air) was repeated for 4 times and 2nd, 3rd, 4th data were averaged as represented data (1st cycle data was discard, because we would like to most stable data). Tish detail was written in 3.1 Analytical performance.

We have already known the stability of the standard gas in 10 L aluminum cylinders from the many experiences in our other long-term observation. We added the data for the stability of the standard gases used for the Mt. Fuji.

10. Pag 3, line 16. You cannot use this comparison of data for evaluating the precision (i.e. the repeatability) of your measurements! I also have serious concern that you can provide an assessment of spatial representativeness by using this data-set. . .

>Yes. It was analytically incorrect to use CONTRAIL data for evaluation of repeatability. We added our comparison work using bottle sampling to assess the accuracy of our measurement system. The results also showed a good agreement within an analytical precision for bottle sampling and measurement system. This detail was written in 3.4 comparison with the bottle sampling data.

It may be difficult to compare our data with CONTRAIL data directly. But as a result, their good agreement in concentration suggested that data at Mt. Fuji could show the similar CO2 concentration level at the altitude over 3600m around Tokai-Kanto district.

11. I think that the authors should spent more work in presenting an assessment of the quality of the system performance. Along the manuscript, the authors provided a statistical analysis (Fig. 5 of the manuscript) of the standard deviation of the measured value of standard gases. This information can be useful to assess the "long-term repeatability" of the measurements (see e.g., Lebegue et al., AMT, 2016). However, to better exploit this point, I would like to see this analysis also split as a function of the different standard gases used and/or as a function of the different years. It should be interesting to see the absolute difference between the $CO_2$ measured values of the standard gases and the reference standard values. Moreover I would like to see information about the uncertainty or expanded uncertainty of the measurements achievable by this system, which is an important parameter missed in the paper.

>We added Table 1 which showed the $CO_2$ first calibrated values of the working standard gas and the lastly calibrated values after use. The differences between them were about 0.1 ppm, means that they are very stable in the cylinders. Also, we can correct this kind of small deviation in the standard gas values by using weighted average with time (even so maybe small uncertainty (30% of 0.1 ppm = 0.03ppm = detection limit of NDIR in NIES lab) will be left). If we looked at the linearity of the calibration curve measured by the system, the deviation from the calibration curve was also small (at most 0.05ppm). Real signal (10 s average) had 0.05ppm on average as a standard deviation for repeating measurement. It is also said that NIES $CO_2$ standard series (NIES09 standard) matched with NOAA (WMO) standard within 0.09ppm.

Therefore, roughly we can estimate uncertainty for our measurement,

$((0.03)^2 + (0.05)^2 + (0.05)^2 + (0.09)^2)^{0.5} = 0.12$ ppm (k=1)

If k=2 (expanded uncertainty), it should be 0.24 ppm.

If we remove the difference between NIES scale and NOAA scale, it is 0.08 ppm (k=1) or 0.16ppm (k=2)

[Figure]

12. Concerning the difference with MLO, which may be the role of using two calibration scales (NOAA vs NIES09)? Please quantify. . . How many often did you re-calibrate the working standard against your National reference (I suppose once per year, but this should be clearly stated in the paper)?

   >We wrote the results of the 6th WMO/IAEA Round Robin inter-comparison whose results that the NIES09 scale was lower than the NOAA scale by 0.04-0.09 in a range of 376-404 ppm. We re-calibrate the working standard gas every two or three years. We created the table 1 which showed the information of standard gas (Days of calibration and re-calibration days of each cylinders).

| Cylinder No | Before the cylinders installation | | | | After the cylinders replacement | | | | Change of concentration | |
| | Date of calibration | Calibrated value (ppm) | Pressure of Cylinder (MPa) | Date of installation | Date of replacement | Date of re-calibration | Re-calibrated value (ppm) | Pressure of Cylinder (MPa) | Change amount of the concentration (ppm) | Change rate (ppm year$^{-1}$) |
|---|---|---|---|---|---|---|---|---|---|---|
| CPC-00449 | 17-Jun-2009 | 368.86 | 10.8 | 16-Jul-2009 | 24-Jul-2011 | 19-Aug-2011 | 368.82 | 3.4 | -0.03 | -0.02 |
| CPC-00447 | 17-Jun-2009 | 383.10 | 11.0 | 16-Jul-2009 | 24-Jul-2011 | 19-Aug-2011 | 383.24 | 4.0 | 0.14 | 0.06 |
| CPC-00448 | 17-Jun-2009 | 403.45 | 10.8 | 16-Jul-2009 | 24-Jul-2011 | 19-Aug-2011 | 403.54 | 3.2 | 0.09 | 0.04 |
| CPC-00445 | 10-Jun-2011 | 367.94 | 11.4 | 25-Jul-2011 | 25-Jul-2013 | 6-Aug-2013 | 368.02 | 0.8 | 0.09 | 0.04 |
| CPC-00450 | 10-Jun-2011 | 383.46 | 11.5 | 25-Jul-2011 | 25-Jul-2013 | 6-Aug-2013 | 383.42 | 3.6 | -0.04 | -0.02 |
| CPC-00451 | 10-Jun-2011 | 402.29 | 11.5 | 25-Jul-2011 | 25-Jul-2013 | 6-Aug-2013 | 402.37 | 3.4 | 0.08 | 0.04 |
| CPC-00043 | 23-Jun-2013 | 367.10 | 12.8 | 26-Jul-2013 | 1-Jul-2016 | 10-Jul-2016 | 367.12 | 2.5 | 0.02 | 0.01 |
| CPC-00448 | 23-Jun-2013 | 393.17 | 12.6 | 26-Jul-2013 | 1-Jul-2016 | 10-Jul-2016 | 393.12 | 2.5 | -0.05 | -0.02 |
| CPC-00449 | 23-Jun-2013 | 418.59 | 12.8 | 26-Jul-2013 | 1-Jul-2016 | 10-Jul-2016 | 418.44 | 2.5 | -0.15 | -0.05 |
| CPC-00445 | 15-Jun-2016 | 389.18 | 13.2 | 2-Jul-2016 | | | | | | |
| CPC-00450 | 15-Jun-2016 | 409.15 | 13.2 | 2-Jul-2016 | | | | | | |
| CPC-00451 | 15-Jun-2016 | 429.16 | 13.2 | 2-Jul-2016 | | | | | | |

13. Another important information to provide is the data coverage (%) over the measurement period: I feel this is rather high, but this represent a parameter (you must mention it along the abstract!) to evaluate the success of the presented system.

   >We got $CO_2$ daily data of 2219 days from July 2009 to December 2015 (for 2354 days), which covered 94% for the observation period. We wrote that on the abstract and results.

14. Pag 6, line 2: "The precision. . . ..is calculated to be about 0.3ppm". As recommended by the WMO/GAW glossary, "precision" is a term to be used only in relative terms. How did you calculate this "precision"? If this is a standard

deviation, it is the "repeatability" or the "reproducibility". Any information about the "uncertainty" of the measurement?

>It was repeatability. We changed an expression from precision to repeatability.

15. Pag 7, line 24. For year 2012 it seems that the CO2 variability (error bars) experienced a diurnal behavior (higher variability during night-time): please comments. Does a diurnal cycle exist for internal temperature or other system parameters that can affect the measurements? Figure 8: you did not provide explanation in the Figure caption for the error bars (please, do it). To investigate if statistical significant diurnal variability exist, you should plot for each hour an average mean value with the 95% confidence interval (not simply the standard deviation).

> In data of 2012, a fairly small lower tendency (lower than 0.4ppm) in concentration may be admitted before noon time. It often included vegetation effect. However, it was concluded that the timing of the air sampling over Mt Fuji did not appear to affect the monthly and yearly averages. We changed an expression of the figure 8. We plot each hour an average mean value with the 95% confidence interval.

[Figure]

16. Pag 8, line 6: ". . .a strong wind always blow". Too generic. Please provide data or references.

>We add the sentence which is annual average wind speed is about 12 m$^{-1}$.

17. Pag 8, line 11: "it was evident that. . .often see in July". It is difficult from this plot to obtain information about monthly values. I would suggest to replace this plot with monthly mean values and related error bars (representing the associated 95% confidence interval). This would also allow to skip Table 1.

>If the CO2 data of Mt. Fuji change from daily data to monthly data, extremely low concentration and high concentration of the CO2 data of Mt Fuji cannot be seen. These specific events are characteristic in the CO2 concentration of Mt. Fuji. We remained the figure of daily $CO_2$ data of Mt. Fuji. Table 1 as monthly average was transferred to supplementary material to skip it.

18. Pag 8, line 24: Since the CO2 is characterized by a long atmospheric lifetime, 72h back-trajectories can explain just a

fraction of its variability. Please comment in the paper. It is possible to plot the back-trajectories (as "number concentration field" or as centroids of the detected clusters) over the spatial domain in Figure 11? In any case, consider to move Figure 11 to SM. You should also provide the percentage of occurrence for the different air mass transport class. The discussion about results reported by Figure 12 is too qualitative. For the summer, it seems that a large amount of variability affect the data set. For each class of air mass transport please provide average values and 95% confidence interval. How did you define background air (Pag 8,line 27)?

>We added some explanation about meaning of 72 hr trajectory. To evaluate the variation of frequency of each sector, we added one Figure about that in the Fig 11. According to the variation of their frequencies we could see some reason for the variation of the concentration, especially in the case of summer. So, we rewrote this section and omitted a large part (e.g. long-term trend). This detail was written 3.5 effect of air mass origin on seasonal variation.

However, we left explanation about relative change in concentration depending on trajectory in each season.

[Figure]

19. Pag 9, line 1: In my opinion, a too low number of Pacific Ocean air-masses were observed during winter (at least basing on the visual inspection of Figure 12 since no information about their seasonal occurrences were provided) for provide any kind of comment. "air from China SOMETIMES. . .". Too qualitative, please be more specific! How many times (frequency) did you observed these spikes with air-mass from China? The sentence from line 9 to 12 is not clear. I think some words were missed somewhere. . .However, I do not think that this paragraph and figure 13 add really important information to the paper, thus they can be skipped.

> We showed the air mass origin in each season with figure that is showed the percentage of occurrence for the different air mass transport in winter. We added numbers in the explanation (e.g. frequency) if it is needed. However, we rewrote this section, based on the suggestion.

20. The comparison with CONTRAILS observations must be better commented (or even skipped). The authors claim for a good agreement but I do not completely agree: a rather large "average" bias (2 ppm) can be deduced by the linear correlation result. In general I do not think that you can use this comparison for "verify" the precision of the measurement system (the term "precision" is not correct at page 10, line 16)! Also concerning the assessment of the representativeness of the measurement site, I think that this analysis can provide only a "qualitative" information (also considering that CONTRAILS observations are not uniformly distributed on the geographical domain). As an instance,

the most part of the CONTRAILS observations are recorded not far from Tokyo. Would this introduce a bias on the contrails data-set due to the urban emissions? Can this (at least) partially explain the 2 ppm bias? Moreover, it is not clear what do you mean for "daily data". For CONTRAILS what kind of daily average did you consider? 24h mean values? Or did you select the hourly data on a time window in agreement with Fuji measurements (i.e. 14:100-17:28 and 21:00-00:28)?

>We changed explanation about this section. Bottle sampling data was included in the section for showing analytical performance. However, CONTRAIL data comparison was important to show the representativeness of measurement at Mt. Fuji.

The bias between Mt. Fuji and CONTRAIL is 0.05 ppm. Two ppm is standard deviation between Mt Fuji and CONTRAIL. The 2 ppm might be originated by the difference in measuring time (Mt. Fuji were taken at night and CONTRAIL's were taken throughout the day).and the measuring places (CONTRAIL data included many cases over Boso peninsula (Chiba prefecture), because Narita airport is located in Chiba prefecture. Some data originated from flights over Nagoya Chyubu airport ). However, data of CONTRAIL at altitude of 3.6- 3.9 km was found to be unaffected by anthropogenic emission directory from Tokyo area (Shirai et al., 2012 at Tellus B), rather affected long-range transport or background air. In this case we are looking at the similarity in CO2 concentration level over these areas.

21. Fig. 14 is almost unreadable. Mt. Fuji points completely overlap the "reference" time series of MLO. I'm wondering if it's really necessary to show the whole long term time series of MLO: for the most part of it, you do not have any data to overlap!!! I would suggest to start the x-axis since 2010 and make the MLO point more visible.

    >We re- arranged the figure to expand it from 2010 to 2015.

22. Fig. 9, line 17: "The RANGE of the annual rates of increase during 2009-2015 was 1.5-2.7 ppm/yr". Actually, the variability in the growth rate can be also related to other factors, as an instance ENSO (as mentioned in the following by authors) or to change in fluxes between atmosphere and terrestrial biosphere.

    >Exactly. In this case, we are checking general trend of the trend, comparing with MLO

23. Pag 10, discussion of Fig. 15. Some statistical analysis should be provided for the linear correlation (i.e. statistical significance). In general, I think that is dangerous to compare and comment (small) long-term CO2 differences between two measurement sites without providing an assessment of the measurement compatibility. . .I'm wondering if these tendency in long-term differences are evident also at other measurement sites. In this case, this would reinforce your conclusions. I recommend to consider open-access data-sets available at the World Data Center for Greenhouse Gases by WMO/GAW and to repeat a similar analysis for other sites in the East Asia/Pacific regions vs MLO.

    > We also agree with your suggestion for this case. The discussion for trend was omitted. We deleted Fig. 15.

TECHNICALS:

24. Pag 2 line 10: "(CDIAC, Boden et al., 2015)" line 15: "air mass transport" line 23: "Mt Fuji is positioned...of the Asian continent". Please provide references or show back-trajectory analysis.

    >We changed from CDIAC, Boden et al., 2015 to CDIAC 00001_V2016; Boden et al., 2015. We added the reference as (Igarashi et al., 2004)

25. Line 22: "Mt. Fuji is located. . ." Line 27: What do you mean for "irregular"? Please use a more specific terminology.

    >We deleted "irregular"

26. Pag3, Line 28, point (v): it is difficult to obtain information on the annual rate of increase basing on so "sparse" measurement.

    >Aircraft measurements which is showed the measurement results at the altitude of around Mt. Fuji (3776 m). We changed the sentence "(v) The observed annual rate of increase was comparable with the rate derived from aircraft measurements over Japan at the equivalent altitude the summit."
    In general, if even we have only data once a week, we can discuss trend roughly if the data shows regional background data.

27. Pag 4 Line 2: "shows a picture. . ." Line 3: "The system consists. . ." Line 16: "In addition, WHEN the temperature. . ."

    > We changed this sentence as you indicated.

28. Pag 8 Line 27:" It was found,. . .IN SUMMER"

    > We changed this sentence as you indicated.

---

## Author Comment (AC2) · 26 Dec 2016

**Response to reviewer #2 Recent six-year atmospheric $CO_2$ concentration at the summit of Mt. Fuji observed by a battery-powered $CO_2$ measurement system**

Shohei Nomura[a], Hitoshi Mukai[a], Yukio Terao[a], Toshinobu Machida[a] and Yukihiro Nojiri[a,b]

[a]Center for Global Environmental Research, National Institute for Environmental Studies, 16-2 Onogawa, Tsukuba, Ibaraki, 305-8506, Japan.

[b]Hirosaki University, Bunkyo-1, Hirosaki, Aomori, 036-8560, Japan.

General comments:

1.  The manuscript presents a technical description of a CO2 monitoring system operated for 6 years at the high altitude site Mt. Fuji, Japan, and associated data analysis. The description of the measurement setup is rather short for a publication in Atmospheric Measurement Techniques and should be extended (see specific comments below). On the other hand, the data discussion could be shortened as it is not the main scope of a paper in AMT. This would be easily possible as the current manuscript extensively compares the Mt. Fuji data with data from Mauna Loa, Hawaii (differences of daily averages (Fig. 10), differences of daily averages for summer and winter (Fig. 12), trends and growth rates for Mt. Fuji and Mauna Loa (Fig. 13), Mt. Fuji and Mauna Loa time series (Fig. 14), trends of monthly Mt. Fuji-Mauna Loa differences (Fig. 15)). Alternatively, the authors can also envisage to elaborate the data interpretation section, e.g. by also incorporating observations from other (high altitude and/or background) monitoring stations, and can consider submitting the manuscript to Atmospheric Chemistry and Physics. Thus, I recommend the publication of the manuscript in AMT after strengthening the experimental section and concisely abridging the data interpretation. A revision of the manuscript with a stronger focus on the data analysis and a submission to ACP seems to be a suitable alternative, too.

    >We focused on submitting to AMT. So, we added our analytical explanation (calibration information, data handling, standard gases handling and stability, our scale and NOAA scale) to strengthen the experimental section. We also deleted some figures and explanation in the section of results to shorten the text, as suggested.

2.  The manuscript contains quite a lot of tables and figures, some of them are to my mind not really needed or can be merged. The authors may consider reducing the number of tables and figures, see also my comments below.

    > We decreased number of figures to six.
    Merged: Figure 1 and 11, and Figure 3 and 4, and Figure 9 and 13,
    Created: one table and two figure, Move to supplementary material; Table 1).

3.  I strongly suggest revising the conclusions which are currently only a summary of what was said before. Please add the lessons-learnt (especially on the instrumental side) – e.g. what would you do differently when you could start from scratch again – and provide an outlook, e.g. are the measurements ongoing? If so, are there any changes/improvements on the measurement setup planned? Did the authors modify the measurement setup during the 6 years of operation, i.e. did they improve their system based on the experience gained in the earlier years?

    >We rewrote the conclusion, according to your suggestion.
    First, we met the difficulty in sending data by ORBCOM satellite. It was so important that we changed the system to Iridium satellite, quickly. We wrote some improvements for measurement and operations such as method for charging 100 batteries. After we installed the system, in 2010 (next year), we developed and installed auto battery charger (Switch for power mode and charge mode). In 2012, we also added the switch box for winter mode and summer mode. For preventing electrical shock from lightning which often happened at the summit we strengthen the earth connection. In 2016, we replaced 50 batteries.

4.  Even if it is common to use expressions such as "the CO2 concentration was 400 ppm" colloquially, it is an incorrect statement and not suitable to be used in the scientific literature. Quantities given in ppm refer to mole fractions or mixing ratios and cannot be called concentrations. Please use the correct terminology throughout the manuscript.

    > We tried to change the word "concentration" to mole fraction.

Specific comments:
Title:
5.  I suggest changing the title to "Six-years of atmospheric CO2 observations at Mt. Fuji recorded with a battery-powered measurement system"

    > We modified this sentence as you indicated.

Abstract:
6.  Lines 15-18: "The difference in monthly average CO2 concentration between Mt. Fuji and MLO appeared to increase from 2009 to 2015. Interannual variability and growth rate of CO2 concentration were similar both at Mt. Fuji and MLO, 13 ppm increase from 2009 to 2015, but the annual average concentration at Mt. Fuji was about 1 ppm higher than at MLO." It sounds like a contradiction (the difference increases but the average was 1 ppm higher), or is at least not understandable without having seen Fig. 15 and the related discussion, respectively. I suggest to delete the first sentence and to add another sentence after the second sentence, like "However, differences in Mt. Fuji and MLO observations show divergent trends depending on seasons."

>We agreed with your advice. But, we decided to shorten the discussion about the trend of CO2 concentration, because our data is still relatively short and it is difficult to clarify the seasonal differences in the trend, which could be influenced by the climatic variation. So, we rewrote only seasonal variation and difference of CO2 concentration by air mass origin about the results of measuring atmospheric CO2 concentration at the summit of Mt. Fuji.

7. Line 18-19: delete "Monthly averaged . . . in April 2013." Not relevant here.

   > We deleted this sentence as you indicated.

8. Lines 21 – 22: change ". . . indicating that Mt. Fuji was a representative site . . . in the mid-latitude Asian region" to ". . . indicating that Mt. Fuji is a representative site to monitor CO2 concentrations in the mid-latitude region."

   >Thank you for the advice. We rewrote abstract. So unfortunately, this part was deleted.

Introduction:

9. Page 3, lines 8-9: ". . . without electricity supply . . .", better say ". . . without gridded electricity supply . . ."

   > We changed this sentence as you indicated.

10. Page 3, lines 8-9: reword sentence: ". . . even under the harsh conditions . . ."

    > We changed this sentence. The explanation was added.

11. Page 3, lines 8-9: " . . . harsh conditions found at the summit of Mt. Fuji, where the atmospheric pressure is low . . ." Why is low pressure a harsh condition?

    > Low pressure is related to the sensitivity of NDIR. Also it is related to our maintenance work at the summit. We added some explanation about harsh condition in introduction.

12. Page 3, line 16: add latitude, longitude and altitude for Mauna Loa

    > We added latitude, longitude and altitude for Mauna Loa.

13. Page 3, line 17: "To evaluate the regional representativeness and precision of the measurements obtained by our system, the data are compared with aircraft observations." Don't you evaluate the accuracy rather than the precision when comparing with other data?

>We agree with your advice. We added another result using bottle sampling data to evaluate the accuracy of our measurement system. Instead, we used comparison with CONTRAIL data to evaluate the regional representativeness of Mt Fuji data. This detail was written in 3.4 comparison with the bottle sampling data.

Methods:

14. This should be the main part of the paper and thus, needs elaboration. Page 3, lines 20-25: add Mount Fuji altitude. How can you access the station? Is access only possible in July and August, or also during the rest of the year (in exceptional cases, e.g. for trouble-shooting).

    >We added altitude of Mt. Fuji. It is 3776 m. Also we added explanation about transportation and how to access the station. Actually, we can access only July and August by using the bulldozer with our goods for exchange. The bulldozer is specialized only for the transportation for public maintenance around Mt Fuji. The operation of the bulldozer was done in only July and August because the operation in September to the following June was interrupted due to snowfall. If CO2 observation is interrupted in September to the following June, we have to wait coming summer to fix the system. We have two set of the main measurement system. So, if we have some trouble, we can exchange them during summer time.

15. Page 4, line 6: add number of pumps (4, according to Fig. 3). Why do you need four pumps? Can you redesign the setup using less pumps reducing the power consumption?

    >We use four pumps for sucking outside air, sending room air to NDIR line, sending outside air to NDIR line and flowing dry air, respectively. The air pump for sending room air to NDIR and the air pump for sending outside air to NDIR work alternatively. So usually 3 pumps are working.

16. Page 4, line 7: write ". . . using a Nafion membrane . . .".What is the dew point that you achieve with the setup? Does the drying efficiency change (decrease) with time? Is a 2 liter cartridge of Silica gel sufficient for the Nafion counter flow for 10 months.

    >We changed the cartridge of Silica gel every year, although over 90% of Silica gel was still blue after one year.

17. Page 4, first paragraph: did you apply any modifications to the measurement setup, in particular to the CO2 analyzer? E.g. disconnecting the display or reducing the flow through the NDIR to reduce power consumption.

    >We did not change the measurement setup in $CO_2$ analyzer. We connect the PC to CO2 measurement only summer. We did not connect PC from September to the following June. The system is controlled by the control board and operated automatically.

18. Page 4, line 17: "… a small internal heater was planned to activate . . ." was it only planned or was it also in place? If the latter is true, write "… a small internal heater was implemented to activate . . ."

   > We changed this sentence as you indicated.

19. Page 4, lines 20-24: did you use any inlet filter? If so, how often was it changed?

   > We forgot to add a filter in the figure. Our line has the filter (Swagelok SS-4F-7) which is changed once a year.

[Figure]

20. Paragraph 2.3 Electrical power system: It remains unclear why gridded power is only available in summer. What does exactly change in early July and late August? Is the station permanently staffed in summer? Did you ever try off-grid generated power with wind turbines or solar panels?

   > To avoid an ignition accident by an electric short circuit, the commercial power is not supplied to the station from September to the following June because the worker is not permanently stationed there in September to the following June. Installation of equipment like wind turbines or solar panels outside the station was tightly restricted by law because Mt. Fuji including the station is categorized the National park.

21. Paragraph 2.4 Measurement sequence: How is the measurement system controlled? Which software is used? Elaborate on the instrument maintenance. Can you remotely access the measurement setup, i.e. can you access the computer when the satellite communication is running? E.g. to modify the measurement sequence. Did you never face any serious instrumental failures? There seems to be a longer data gap in 2012 (according to Fig. 10)? What happened? What is the overall data coverage based on your daily averages? What is the yearly consumption of reference gas? How long do the reference cylinders last?

> Measurement is controlled by the control board (MC-mini, Kimoto Electric CO., Ltd.). We can remotely access basic setup parameters when the satellite communication is running (e.g. changing start time). However, we did not use these command because satellite communication was not so smooth. We added some explanation about satellite communication in the text.

Our measurement system got the lightning influence in 2012 and 2014. One board was damaged and exchanged. We got $CO_2$ daily data of 2219 days from July 2009 to December 2015 (for 2354 days), which was covered 94% for the observation period. We wrote that on the abstract.

The working standard gases were consumed 300L per year. We replaced the cylinders once every two or three years. We added Table 1 for information of working standard gas, as suggested.

Table 1. The information of working standard gas in observation period

| Cylinder No | Before the cylinders installation | | | | After the cylinders replacement | | | | Change of concentration | |
| | Date of calibration | Calibrated value (ppm) | Pressure of Cylinder (MPa) | Date of installation | Date of replacement | Date of re-calibration | Re-calibrated value (ppm) | Pressure of Cylinder (MPa) | Change amount of the concentration (ppm) | Change rate (ppm year$^{-1}$) |
|---|---|---|---|---|---|---|---|---|---|---|
| CPC-00449 | 17-Jun-2009 | 368.86 | 10.8 | 16-Jul-2009 | 24-Jul-2011 | 19-Aug-2011 | 368.82 | 3.4 | -0.03 | -0.016 |
| CPC-00447 | 17-Jun-2009 | 383.10 | 11.0 | 16-Jul-2009 | 24-Jul-2011 | 19-Aug-2011 | 383.24 | 4.0 | 0.14 | 0.065 |
| CPC-00448 | 17-Jun-2009 | 403.45 | 10.8 | 16-Jul-2009 | 24-Jul-2011 | 19-Aug-2011 | 403.54 | 3.2 | 0.09 | 0.042 |
| CPC-00445 | 10-Jun-2011 | 367.94 | 11.4 | 25-Jul-2011 | 25-Jul-2013 | 6-Aug-2013 | 368.02 | 0.8 | 0.09 | 0.041 |
| CPC-00450 | 10-Jun-2011 | 383.46 | 11.5 | 25-Jul-2011 | 25-Jul-2013 | 6-Aug-2013 | 383.42 | 3.6 | -0.04 | -0.020 |
| CPC-00451 | 10-Jun-2011 | 402.29 | 11.5 | 25-Jul-2011 | 25-Jul-2013 | 6-Aug-2013 | 402.37 | 3.4 | 0.08 | 0.036 |
| CPC-00043 | 23-Jun-2013 | 367.10 | 12.8 | 26-Jul-2013 | 1-Jul-2016 | 10-Jul-2016 | 367.12 | 2.5 | 0.02 | 0.008 |
| CPC-00448 | 23-Jun-2013 | 393.17 | 12.6 | 26-Jul-2013 | 1-Jul-2016 | 10-Jul-2016 | 393.12 | 2.5 | -0.05 | -0.016 |
| CPC-00449 | 23-Jun-2013 | 418.59 | 12.8 | 26-Jul-2013 | 1-Jul-2016 | 10-Jul-2016 | 418.44 | 2.5 | -0.15 | -0.050 |
| CPC-00445 | 15-Jun-2016 | 389.18 | 13.2 | 2-Jul-2016 | | | | | | |
| CPC-00450 | 15-Jun-2016 | 409.15 | 13.2 | 2-Jul-2016 | | | | | | |
| CPC-00451 | 15-Jun-2016 | 429.16 | 13.2 | 2-Jul-2016 | | | | | | |

22. Page 5, lines 15-18: "However, we subsequently changed the operational time to 21:00–00:28 JST, to avoid local daytime influences from transportation of the air mass around Mt. Fuji that might affect the $CO_2$ concentration over the summit of Mt. Fuji, which is similar to how observations are obtained at MLO." To my knowledge, $CO_2$ measurements at MLO are continuous and a filter to extract background conditions is applied afterwards. However, most background data are identified at Mauna Loa in the late afternoon, see http://www.esrl.noaa.gov/gmd/ccgg/about/co2_measurements.html for more details.

>Yes. We first though that mid latitude had higher wind velocity than sub-tropical area in Hawaii. Therefore, we took usual daytime for sampling. However, we were afraid that daytime concentration might be influenced by local $CO_2$ fluxes in some cases. After we checked daily variation, we found that daytime sampling looked ok but nighttime seemed better for sampling, as written in the text.

23. Page 5, line 22: did you use an inlet filter when measuring room air?

>We forgot to add an inlet filter of room air in Figure. We added the filter to Figure.

[Figure]

24. Page 5, line 22: ". . . to stabilize the flow line." What does that mean?

> Outside air is sucking by one air pump. It will take some time for the pump to purge the PTFE line from air inlet. During that period, room air is introduced in the NDIR line and also dry air started to circulate to familiarize inside the tube with the fresh air. After that, outside air is introduced and measured.

25. Page 5, lines 23-24: Why do you need to push the air into the analyzer when another pump sits behind?

>This pump is used for sucking air with a higher flow rate (1.5L/min), because we have to purge air in PTFE tube quickly and minimize contamination from the tube. Also, because in winter air inlet may get some moisture frozen, we need a kind of powerful pump to introduced the air from the outside. But other pumps are rather small pump which flow the air by 50 ml/min.

26. Page 5, line 27: why does it need one hour to send the data? How large are the data files? Which time resolution do you store the CO2 data?

>The file size is 192 byte. We added explanation in the text. The communication become difficult under bad weather conditions like the summit covered the cloud. So that we prepared one hour for the data communication. Usually, the communication time is finished just 1 or 2 minutes. But it sometime failed.  After the we changed the system from ORBCOM to Iridium, the situation became better. Within 1 hr time window, the system tried to communicate if the system has any data, which could not sent last time, in addition to the new data.

27. Page 5, lines 28-29: "The derived concentration was based on the average of the data from the second, third, and fourth cycles . . .", in other words, the daily average is based on 3 x 8 min = 24 minutes of observations. In fact, it will be even less since you have to discard some data to account for flushing and

signal stabilization after switching from room air to outside air. Add the information how many minutes of data were discarded after switching. Please mention explicitly that the daily averages used below only rely on a very short measurement period.

> We added the explanation about that.  The daily average is based on 3 x 6 min data (because we discarded the first 2 minutes data for 8 minutes data) = 18 minutes of observations.".

28. Paragraph 2.5 Continuous measurements in summer: Does the continuous system use the same inlet? Did you use the same reference gases? Does the default system operated all year long also only measure for 3.5 hours a day in summer? The summer system has no dryer. Did you test and quantify the $CO_2$ losses in the Nafion dryer?

>The continuous system used the same inlet as the battery-powered measurement. The continuous system used other working standard gas (But same scale (NIES09) as the battery-powered measurement). The default system measured $CO_2$ concentration 4 times per day in summer of 2010-2012 to compare bottle sampling measurement. But that, the system worked for 3.5 hours a day.
   We checked the measurement of the system without dryer, and found that measurement data matched with our system within 0.3ppm and stability was even better than that.  So we only used this system to detect daily variation of $CO_2$ conc at Mt. Fuji.
   As for Nafion effect, because we used the same outside air to dry the sample air, the change of $CO_2$ conc must be negligibly small.

29. Paragraph 2.6 Weather data: How does the power supply for the meteorological measurements look like?

>The temperature inside box and room were measured by our system. We forgot to add the sensor of temperature in Figure. We added the sensor to Figure. However, JMA measured outside temperature by their own batteries system. But it must be small size. We added some explanation about it in 2.7 weather data.

(b)

[Figure]

30. Page 7, lines 14-19: again, didn't you experience any other interruptions, in particular during the 10 months of unattended operation?

>The interruptions occurred only twice in observation period. It was occurred by lightning (April 2-July 23, 2012 and August 1-18, 2014). We added explanation in 3.2 operation with 100 batteries over 6 years.

31. Paragraph 3.2 and onwards: Use data other than Mauna Loa for comparison and interpretation. E.g. Lulin (Taiwan), Niwot Ridge (USA), Mt. Waliguan (China) etc.; use the marine boundary layer reference (available at http://www.esrl.noaa.gov/gmd/ccgg/mbl/) for comparison which is also available for the Mt. Fuji latitude. How does the difference in latitude (Mt. Fuji – Mauna Loa) influence your comparison?

>Although data from Waliguan is limited, we had some comparison with it. At the lower concentration event in summer, the concentration was comparable with each other. But the data in Waliguan was changed by some reason, at present it is difficult to use it. Mauna Loa showed average of CO2 concentration of mid-latitude in Northern hemisphere. So, it can be used for a typical background CO2. Lulin is located in rather southern part, which is close to our Hateruma station. Hateruma data shows a similar feature to Mt. Fuji in seasonal pattern. To analyze CO2 trend around East Asia, the comparison between Fuji and MLO seemed better.

32. Page 7, line 29: add altitudes for Niwot Ridge and Hakkouda

>We added the altitudes for Niwot Ridge and Hakkoda. Niwot Ridge in the U.S.A. (40.05°N, 105.59°W,

3528 m a.s.l.), and Hakkoda (40.41°N, 140.51°E, 1324 m a.s.l.)

33. Page 8, lines 4 – 8: move this paragraph up to paragraph 2.1

> We moved this paragraph to 2.1.

34. Page 8, line 13: say "18 ppm larger", add amplitudes for Mt. Fuji and MLO.

>We added amplitude for MLO. It is 8 ppm.

35. Page 8, line 16: Table 1 is not needed, in particular if data are available in a publicly accessible data repository. Did you submit the data to the World Data Centre for Greenhouse Gases?

>We did not submit our $CO_2$ data to WDCGG yet. But we will release the data on our website and submit to WDCGG after this paper is accepted to the journal. We moved Table 1 to supplementary material.

36. Page 8, lines 22 – 23: "Conversely, the CO2 concentration from December to March at Mt. Fuji was generally higher than at MLO." February and March are the moths with most intense biomass burning on the Indochinese Peninsula. Do these fires affect the observations at Mt. Fuji?

> We think that the CO2 data of Mt. Fuji is also affected by biomass burning on the Indochinese Peninsula. We added such possibility in the text.

37. Page 9, line 5: add reference to Fig. 1

>We added the reference of (Watanabe et al., 2000)

38. Page 9, lines 12-13: this statement is based on the Fourier-transformed (i.e deseasonalized) data, correct?

>Yes

39. Page 9, lines 20- 21: ". . .the increased rate of growth of CO2 concentration because of accelerated plant respiration over land and weakened photosynthesis activity". Add reference. Next to vegetation effects, it is also due to more intense biomass burning, see e.g. Betts et al., Nature Climate Change, September 2016).

>Thank you for giving information. We deleted this paragraph, but added about biomass effect at the section of seasonal variation.

40. Page 9, lines 24-26: "For example, the negative values of ΔCO2 concentration have enlarged gradually over the six–year period of 2009–2014, as shown in Fig. 10. Chen et al. (2014) reported that growth of Asian vegetation increased recently." I doubt that such an effect can be seen in a 6-year time series. Moreover, Chen et al. refer to changes over the last three decades.

>We deleted this paragraph because we have only six-year data of Mt. Fuji. It is difficult for discussing the trend of CO2 concentration.

41. Page 9, lines 26-28: "In particular, remote sensing observations over eastern Siberia have revealed a notable increase in vegetation during recent decades. Correspondingly, the negative values of ΔCO2 concentration have enhanced gradually." Is there any proof confirming this statement. How about CO2 observations at other Asian sites? Can't it also be an emission effect with decreasing emissions in summer?

>Yonaguni site in Japan near Taiwan is also decreasing ΔCO2 concentration in summer. But we quit to discuss this phenomenon on this paper because we have only six-year data of Mt. Fuji. Also in our observing period, we admitted some annual variation in trajectory sector fraction from Asian Continent in summer, suggesting that the variation would give more influence to such events.

42. Page 9, lines 28-29 and page 10, lines 6-7: "This phenomenon might be related to the increase of anthropogenic CO2 emissions in China during recent decades" and". . .the positive trend in winter was not so significant, which might be attributable to the slowing of the growth rate of CO2 emissions in China during 2011–2014." Sounds like a contradiction.

>We deleted this paragraph. The increase of anthropogenic CO2 emissions in China increased until 2010. But recently (2010-2015) the increase rate is decreasing.

43. Page 9, lines 20-32: "Such event-based phenomena should be evaluated by numerical simulation, but we expected that such signals from regional emissions or absorption changes would be included in the data at Mt. Fuji." If you expected it, why didn't you look closer at it?

> We deleted this paragraph. But we looked at it a little closer using trajectory frequency.

Conclusions:
44. As mentioned above, revise the conclusions and add lessons-learnt and an outlook. Tables and Figures: Table 1: not needed, a release of the data in a public data repository is strongly encouraged.

>We revise the conclusion. We moved Table 1 to supplementary material and we will release our CO2 data by our DOI system later.

45. Table 2: not needed, some numbers could be incorporated in Fig. 15.

   >We deleted Table 2

46. Fig. 1: add areas of air mass origin (Fig. 11) to Fig. 1 and delete Fig. 11.

   >We merged Fig 11 with 1.

[Figure]

47. Fig. 5: what is the nominal value of the standard gas?

   > We wrote those values in Table 1.

Table 1. The information of working standard gas in observation period

| Cylinder No | Before the cylinders installation | | | | After the cylinders replacement | | | | Change of concentration | |
| | Date of calibration | Calibrated value (ppm) | Pressure of Cylinder (MPa) | Date of installation | Date of replacement | Date of re-calibration | Re-calibrated value (ppm) | Pressure of Cylinder (MPa) | Change amount of the concentration (ppm) | Change rate (ppm year$^{-1}$) |
| --- | --- | --- | --- | --- | --- | --- | --- | --- | --- | --- |
| CPC-00449 | 17-Jun-2009 | 368.86 | 10.8 | 16-Jul-2009 | 24-Jul-2011 | 19-Aug-2011 | 368.82 | 3.4 | -0.03 | -0.016 |
| CPC-00447 | 17-Jun-2009 | 383.10 | 11.0 | 16-Jul-2009 | 24-Jul-2011 | 19-Aug-2011 | 383.24 | 4.0 | 0.14 | 0.065 |
| CPC-00448 | 17-Jun-2009 | 403.45 | 10.8 | 16-Jul-2009 | 24-Jul-2011 | 19-Aug-2011 | 403.54 | 3.2 | 0.09 | 0.042 |
| CPC-00445 | 10-Jun-2011 | 367.94 | 11.4 | 25-Jul-2011 | 25-Jul-2013 | 6-Aug-2013 | 368.02 | 0.8 | 0.09 | 0.041 |
| CPC-00450 | 10-Jun-2011 | 383.46 | 11.5 | 25-Jul-2011 | 25-Jul-2013 | 6-Aug-2013 | 383.42 | 3.6 | -0.04 | -0.020 |
| CPC-00451 | 10-Jun-2011 | 402.29 | 11.5 | 25-Jul-2011 | 25-Jul-2013 | 6-Aug-2013 | 402.37 | 3.4 | 0.08 | 0.036 |
| CPC-00043 | 23-Jun-2013 | 367.10 | 12.8 | 26-Jul-2013 | 1-Jul-2016 | 10-Jul-2016 | 367.12 | 2.5 | 0.02 | 0.008 |
| CPC-00448 | 23-Jun-2013 | 393.17 | 12.6 | 26-Jul-2013 | 1-Jul-2016 | 10-Jul-2016 | 393.12 | 2.5 | -0.05 | -0.016 |
| CPC-00449 | 23-Jun-2013 | 418.59 | 12.8 | 26-Jul-2013 | 1-Jul-2016 | 10-Jul-2016 | 418.44 | 2.5 | -0.15 | -0.050 |
| CPC-00445 | 15-Jun-2016 | 389.18 | 13.2 | 2-Jul-2016 | | | | | | |
| CPC-00450 | 15-Jun-2016 | 409.15 | 13.2 | 2-Jul-2016 | | | | | | |
| CPC-00451 | 15-Jun-2016 | 429.16 | 13.2 | 2-Jul-2016 | | | | | | |

48. Fig. 6, caption: Write "Monthly averages of ambient temperatures outside . . .."

   > We deleted the figure of monthly averages of temperatures and created the figure of daily data of temperatures. Also merged temperatures data and voltage data.

[Figure]

49. Fig. 11: merge with Fig. 1 and delete

   >We merged with Fig 11 and 1.

[Figure]

50. Fig. 14: does it show daily averages? Monthly averages? The figure repeats Fig. 9, the long-term evolution is not of real interest here. I suggest to delete it.

   >We deleted Fig. 14

51. Fig. 15: only a few trend lines are plotted: for which months? Use open symbols for the months without trend line?

   >We deleted Fig. 15 to concentrate the explanation of the information of working standard gas and method of measurement.

52. Fig. 16: add information to Fig. 1 and delete Fig. 16

   >We deleted Fig. 16

Minor comments:

53. Page 2, line 11: start with lowercase letter after semicolon.

   >We modified this sentence as you indicated.

54. Page 5, line 12: typo "Phenobaboad"

>We modified this sentence as you indicated.